# Inadvertent Dural Puncture Causing Low Pressure Headache and Peripheral Vestibular Bilateral Damage

**DOI:** 10.3390/audiolres15020018

**Published:** 2025-02-20

**Authors:** N. S. Longridge, A. I. Mallinson, R. G. Robinson

**Affiliations:** 1Division of Otolaryngology, Department of Surgery, Faculty of Medicine, University of British Columbia, Vancouver, BC V6T 1Z4, Canada; nslongridge@hotmail.com; 2Division of Neurology, Faculty of Medicine, University of British Columbia, Vancouver, BC V6T 1Z4, Canada; gordon.robinson@vch.ca

**Keywords:** otoliths, vertigo, dural puncture, VEMPs

## Abstract

Introduction: This edition of the *Audiology Research* journal is dedicated to Dr. John Epley. For this reason, we would like to present a very unusual case involving the development of a bilateral abnormality of the otolith organs. The otoliths are the structures from which calcium particles detach to induce benign paroxysmal positional vertigo, the disorder for which Dr. Epley is famous for managing. Case report: This case report outlines an unusual presentation of vestibular pathology involving the otoliths. Discussion: We suggest that the atypical presentation may be related to a bilaterally asymmetrical insult to the otoliths. Conclusions: The central insult suffered by this patient resulted in bilateral peripheral otolithic pathology.

## 1. Introduction

Benign paroxysmal positional vertigo (BPPV), the most common and best understood vestibular disorder, was outlined and described by Dr. John Epley. He developed, explained and popularized his eponymous maneuver, which is now utilized by clinicians worldwide as both a diagnostic and therapeutic tool. It should be kept in mind by diagnostic clinicians that BPPV by definition is an indicator of otolithic pathology, and this case report is an outline of an atypical otolithic presentation.

Epidural steroid injections can sometimes inadvertently result in a dural puncture, causing intracranial hypotension. A rare complication of this can be acute vertigo. When this occurs, it is assumed to be of central nervous system origin. An extensive review of this topic [1] mentioned one reference of “vertigo” as a symptom but their use of the word vertigo was unclear, as the reference cited in that review [2] was somewhat nonspecific, reporting that 60% of the patients in their study complained of “dizziness” and not “vertigo”.

An exhaustive search of the literature by us has turned up only two case reports. The first was a case of “dizziness without headache” after a dural puncture [3]. This patient was diagnosed with “positional vertigo” related to the neurological sequelae of the inadvertent puncture. She was treated with a blood patch after which there was “complete resolution of her symptoms”. The second report was of a patient with spontaneous dural leakage of cerebrospinal fluid with resultant “Meniere-like symptoms” with noted venous engorgement in the internal auditory canal on MRI [4]. That report suggested that the spontaneous intracranial hypotension as a result of the dural puncture was the “curable reason” for the patient’s vestibular symptoms.

Holdgate and Cuthbert [5] also reported that vertigo can be one of the symptoms of low CSF pressure and “stretch of intracranial structures”. We report a patient who suffered persistent peripheral vestibular deficits as a result of an inadvertent dural puncture.

## 2. Case Report

Our patient was the properly restrained driver of a vehicle that was rear ended. She struck her head on the headrest. She had no complaints of hearing loss, tinnitus, dizziness or nausea after the accident. Her main complaint was of moderate pain in the lumbar region, radiating into both legs. These complaints persisted, and she underwent a CT scan which showed bulging disks at L3-4, L4-5 and L5-S1. Because of these findings, a weekly series of Botox epidural steroid injections was commenced 11 months post accident.

On the evening of the third injection, she developed right ear pain and a severe headache which resulted in a visit to the emergency department. Her severe headache was relieved by lying supine and was worsened by standing up. She was diagnosed in the emergency department as having a low-pressure headache. While there, she developed vertigo, nausea, vomiting and prostration, which were made worse by head movement. She had never had vertigo previously. She was also markedly off balance. These complaints improved over a week. Subsequently she had weekly recurrent episodes of spinning lasting 5 to 10 seconds. She had dizziness upon getting out of bed or stooping quickly. She had no other dizziness induced by positional changes (turning over in bed or getting into bed).

After the dizziness began, she noticed that visual stimuli such as busy carpeting and checkered floors made her feel slightly dizzy. When initially seen for ongoing vestibular symptoms two years after the accident, she recalled that two or three months prior to her assessment she had noticed difficulty with depth perception in low light at the movies, and fell a week later because of this, injuring her coccyx.

She was having spells of spontaneous episodes of nausea and vertigo lasting approximately 10 minutes and occurring weekly. In addition, she had to be very careful because of dizziness being induced by being on escalators. This had developed since the onset of her acute vertigo. She was also bothered by vehicles going past her.

A routine ENT examination was normal. The Romberg test with eyes closed was negative, but she swayed during tandem Romberg and tandem walking with eyes closed. A Dix–Hallpike maneuver using Frenzel’s glasses was negative. There was no post-head-shake or post-head-thrust nystagmus. She had no cerebellar or cranial nerve findings.

Audiovestibular investigation was undertaken for the first time 2 years post accident. Pure tone, speech, word recognition and immitance audiometry were all normal. Video Nystagmography (VNG) was normal, including calorics, which simulated the very worst of her vertiginous attacks that had started on the night of the third epidural injection.

Computerized Dynamic Posturography (Sensory Organization Test) was abnormal (Figure 1). Performance was just below the accepted limits for normal on condition 1. Conditions 2 and 3 were normal but conditions 4, 5 and 6 were abnormal. The sensation induced during this test closely simulated her ongoing complaints brought on by movement in her visual environment.

Cervical Vestibular Evoked Myogenic Potentials (CVEMPs) were normal (Figure 2).

Ocular Vestibular Evoked Myogenic Potentials (OVEMPs) were abnormal (Figure 3). Amplitudes were normal bilaterally. There was a delayed n1 and p1 latency on the right, beyond the accepted upper limit for normal. On the left there was an early p1 latency, below the lower limit for normal [6]. There was an abnormal interaural amplitude ratio (IAR).

Subjective Visual Vertical (SVV) results were abnormal. The Head Impulse Test (HIT) suggested some covert saccades in the right posterior plane but was otherwise normal.

At a follow up two years later, the ten-minute episodes of dizziness had settled, but she was still experiencing brief spinning spells lasting about five seconds occurring once a week. They could come on at any time, if she was sitting, standing, walking or lying. The checkered carpet still bothered her. She continued to have dizziness induced by watching traffic going past at a crosswalk and said she almost always crossed a street with somebody else, as she was concerned about falling. She was still careful getting on escalators because of dizziness and continued to have persistent difficulty with foot placement in low light such as in a movie theater.

Her ENT examination was unchanged.

Because of her persistent vestibular symptoms, a vestibular reinvestigation was undertaken 4 years post accident. CVEMPs were now abnormal, with a response amplitude below the accepted lower limit for normal bilaterally. OVEMPs were still abnormal. The interaural amplitude ratio remained high, above the accepted upper limit for normal. However, OVEMP latencies on both sides had returned to the normal range. The HIT was not repeated on her follow up visit.

## 3. Discussion

This patient suffered a dural puncture during her third epidural steroid injection as treatment for lumbar disk protrusions, with resulting characteristic low-pressure headache, and development of coincident acute vestibular symptoms.

It is highly unlikely that the initial motor vehicle accident was the cause of these vestibular symptoms, as they arose 11 months post accident without any dizziness in the post-accident period. It is stated in the literature that symptoms occurring after trauma are not related to the trauma if they began more than six months after the trauma [7]. The vertigo came on immediately after the third epidural steroid injection, following a dural puncture for treatment of her low-pressure headache.

Her acute vertigo suggested either a brainstem insult, (which is highly unlikely because there were no other brainstem complaints or signs), or a peripheral vestibular insult. We hypothesize that she suffered a bilateral but asymmetrical peripheral (inner ear) vestibular insult, as with symmetrical peripheral vestibular insults, vertigo does not occur [8]. This is supported by the initial bilateral VEMP abnormalities. We hypothesize that the vestibular damage resulted from either a mechanical effect of traction on the superior vestibular nerve, or a vascular insult due to traction of vascular loops into the internal auditory canal, with either of these possible mechanisms resulting in damage to the peripheral end organs bilaterally.

The development of these vestibular symptoms involving visually induced dizzy symptoms were initially referred to as VVM. This set of symptoms strongly suggests otolithic involvement [9], and is now referred to in the literature as persistent postural perceptual dizziness, or PPPD [10]. It resulted from the inadvertent dural puncture associated with the low-pressure headache, as her set of symptoms occurred immediately after. It is also unlikely that the measured VEMP abnormalities, which indicate inner ear pathology, occurred as a result of the motor vehicle accident.

There were abnormalities measured on the CVEMP and OVEMP tests. There was variation between the two sets of CVEMP and OVEMP results. The CVEMP assessment was initially normal but turned abnormal two years later. Her OVEMP reassessment showed variations in latencies. At one time it was assumed that variations in VEMP tests meant that the tests were inconsistent and unreliable. However, repetitive testing on a number of other patients who have had ongoing vestibular symptomatology led us to the conclusion that fluctuations in the function of the inner ear balance organs (as shown by these tests) can account for the patient’s fluctuating but persistent symptoms. This can be explained by the fact that an attempt to repair damaged vestibular otolithic membranes results in fluctuating first-order vestibular afferent neuronal activity and fluctuating patient symptoms. VEMP assessment has recently been shown to be very sensitive at detecting bilateral pathology [11] and our assessment was able to document bilateral involvement in this patient. In patients with bilateral vestibular involvement, there is not a normal otolithic functioning side to utilize, which is crucial in the compensation process. As a result, patients’ complaints can be longer lasting and potentially permanent [11].

## 4. Conclusions

This patient suffered a low-pressure headache and an acute peripheral vestibular event after an inadvertent dural puncture. She developed classic otolithic symptoms which were in keeping with the otolithic abnormalities that were documented. We feel it is important to document this case, as it illustrates that the central insult she suffered caused bilateral peripheral vestibular pathology of the inner ear. Unfortunately, her bilateral involvement resulted in persistent symptoms.

## Figures and Tables

**Figure 1 audiolres-15-00018-f001:**
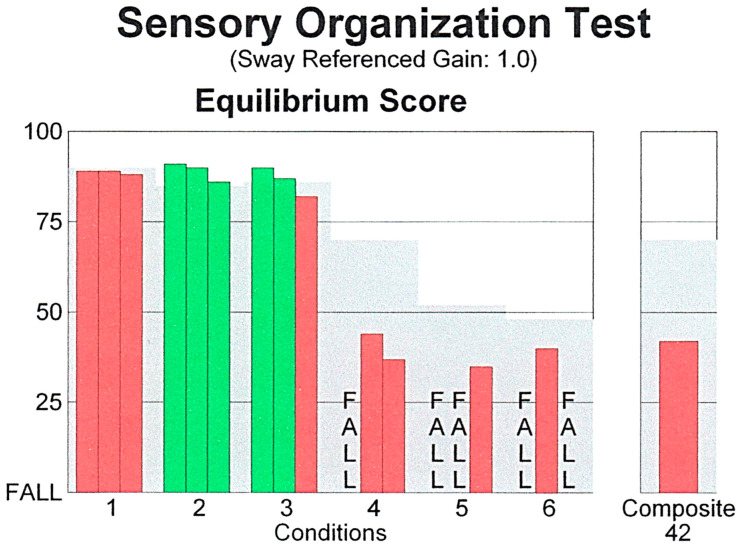
Abnormal posturography suggesting vestibular deficit (Normative data is illustrated by the gray background; performance scores below normal but without falls are shown in red).

**Figure 2 audiolres-15-00018-f002:**
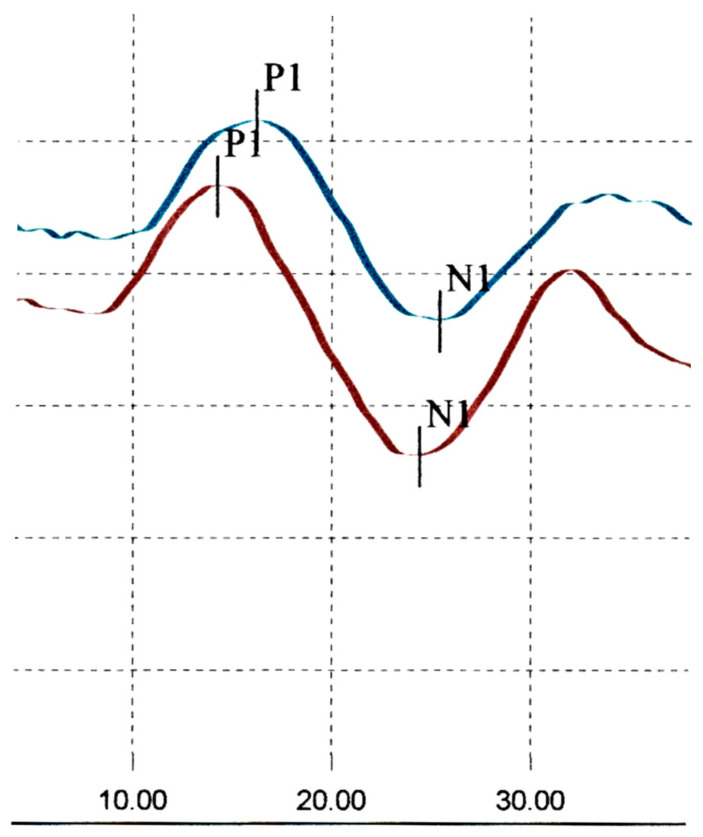
Normal CVEMPs–initial assessment; (right ear responses in red, left ear responses in blue).

**Figure 3 audiolres-15-00018-f003:**
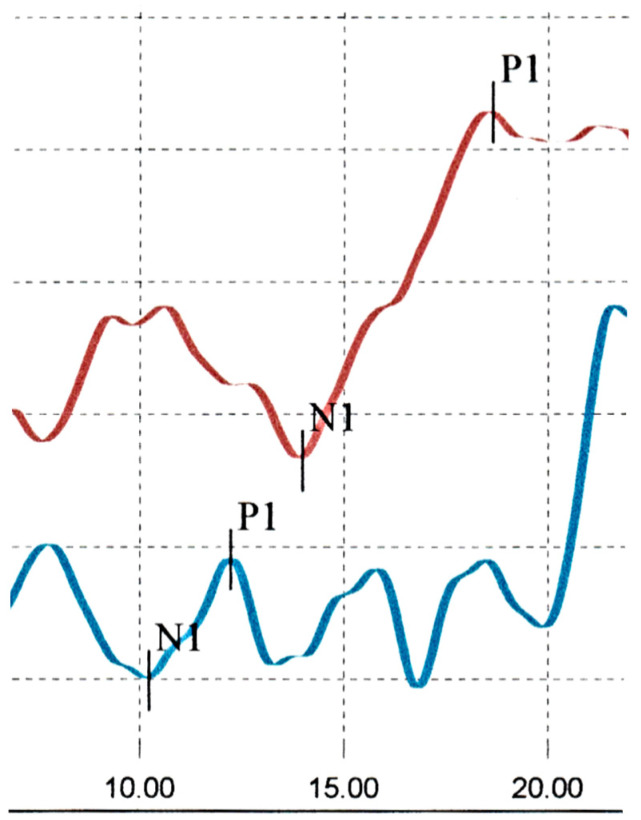
Abnormal OVEMPs–initial assessment; (right ear responses in red, left ear responses in blue).

## Data Availability

Confidential data is held at the hospital facility (Vancouver General Hospital) where all assessments were carried out.

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
