# Peer review of "Inadvertent Dural Puncture Causing Low Pressure Headache and Peripheral Vestibular Bilateral Damage"

_audiolres, 2025, doi:10.3390/audiolres15020018_

Round 1
Reviewer 1 Report
Comments and Suggestions for Authors
Thank you for the opportunity of reviewing your case report. While the findings of acute vertigo following LP/inadvertent LP following an epidural steroid/anesthetic/botox injection are relatively rare (ie compared to a post LP headache) it is a phenomenon that I personally have seen prior and remains somewhat of an enigma for many of us who practice neurotology.
A few questions/comments according to the sections of your manuscript:
1. Introduction Section
a. Lines 22-23: Did your literature review look primarily at inadvertent CSF leakage after an intended epidermal injection and/or did it include those who had also had an intended LP for diagnostic reasons as well (ie for IIH, suspicion of meningitis etc)? I appreciate that the second case however reported in your article (reference 4) did occur on the backdrop of a spontaneous CSF leak.
b. Comment: It is of interest to note that not much has been reported recently in the world literature on this subject matter. One might speculate that reasons could include the relative rarity of this finding or that the majority of patients in whom it occurs have resolution of their symptoms as possible explanations.
2. Care Report Section
a. Lines 63-66: Your patient had recurrent attacks of episodic vertigo lasting approximately 10 minutes without auditory/CNS dysfunction. Would your diagnosis still have been consistent with an otolithic cause or could there have also been angular acceleration receptor involvement additionally? By definition it sounds as if she could also have been diagnosed to have experienced spells of vertigo consistent with a recurrent vestibulopathy (RV) on clinical grounds.
b. Lines 96-101: Your repeat inner ear investigations demonstrated the presence of fluctuation in otolithic activity? Was a repeat VNG ever performed. Was there any consideration at the time whether the phenomenon of hydrops was present? See my questions/comments in the Discussion Section below.
Discussion Section
a. Lines 103-105: While the vestibular symptoms may have been co-incident I concur that they were related to the inadvertent LP resulting in CSF hypotension and not the MVA itself.
b. Lines 108-109: While respectful of Professor Luxon I would like to point out that this is not always the case as there are always exceptions to the rules. One classic exception would be the phenomenon of delayed endolymphatic hydrops (DEH) where attacks of vertigo can occur months, years and even decades following a loss of hearing in the involved ear (even if vertigo were not present at the time of initial loss of hearing related to trauma or non-traumatic events). This is mentioned as there has been speculation that Meniere's disease occurs as a result of injury to the inner ear at some point in time that leads in a delayed fashion to the pathophysiological finding of hydrops in many (but not all archival temporal bones) associated with its classical clinical findings. Hydrops overall can also be a non-specfic finding reflective of trauma to the inner ear for numerous reasons.
c. Lines 112-120: There is certainly much to speculate for the pathophysiological cause of your patient's symptoms. To add to the mix you might also wish to entertain the speculation that a peripheral vestibular loss could have occurred where CSF hypovolemia caused a "vacuum like" effect in the perilymph if there were patent cochlear aqueducts leading to inadvertent inner ear trauma? A further investigation to consider might be that of a high resolution CT scan of the temporal bones to look specifically at the cochlear aqueducts and otic capsules.
d. Lines 121-122: Please define what "VVM" means for the readership.
e. Lines 128-143: As mentioned in the Case Report Section above your findings to date have demonstrated an individual who has fluctuation otolithic activity on VEMP testing and continued problems with the compensation process. A reason that compensation may be incomplete could be the fluctuation in vestibular activity and whether this might be related to a hydropic type presentation. Did you ever consider performing electrocochleography (ECoG) with and without straining, serial frequency threshold cVEMP studies and a delayed contrast gadolinium temporal bone MRI (I acknowledge that this study is not available at all institutions)? You may wish possibly to include these thoughts in your Discussion Section.
Conclusion Section
a. Line 148: As one consideration would you be willing to reconsider whether she actually suffered a "central insult" causing her symptoms (ie injury to the CNS)?
Author Response
Inadvertent CSF leakage with low pressure headache is a well recognized phenomenon. It is unpleasant, but resolves satisfactory in most patients with conservative therapy and, if necessary, dural patching. Associated dizziness is rare as our literature review demonstrated, and ongoing persistent dizziness is even rarer. With any rare disorder, case reports represent the first way that these disorders are reported in the literature and absence of subsequent reports of similar cases or series of cases confirms the rarity of such disorders, which is probably the case here.
As the reviewer notes, the recurrent vertiginous episodes lasting 10 minutes fall within the spectrum of the disorder “recurrent vestibulopathy”. This disorder occurs without headache, so it is unlikely to be due to vestibular migraine. It was described in a series of three papers by Barber and colleagues [1–3] and it was speculated that it might arise on a viral basis in some patients [4]. A follow up study [3] confirmed that this can develop into Menieres disease frequently, but the time sequence here makes this extremely unlikely.
The episodic VVM in this patient developed later, but clearly its development probably indicates that there was damage to the macular system, as the reviewer suggests. The authors agree that this is probably the case in many patients who have vertiginous dizziness, however our ability to investigate this effectively has only been possible since the advent of VEMP testing.
The reviewer correctly notes that fluctuating VEMP test results were recorded, and that repeated caloric testing might have been undertaken. As the reviewer knows, caloric testing is an extremely unpleasant experience for the patient. Vomiting is induced in 10% of patients and severe nausea occurs in 30%. Acute migrainous symptoms may occasionally be induced by caloric testing as well [5]. In the hands of the authors, caloric testing is only undertaken when a detected abnormality will change the patient’s management; for example by using low risk therapy such as intratympanic medication unilaterally, or even surgical intervention, so in this case, the test was not repeated as the disorder was bilateral and therefore not amenable to low risk therapy.
The reviewer is correct that delayed endolymphatic hydrops can follow trauma [6],but in the described case, the patient had never had any significant hearing loss, tinnitus,or episodic vertigo previously in her life and had not sustained any head trauma other than that described in the motor vehicle accident.
The reviewer has put forward a suggestion that the ongoing headache and ongoing symptoms could have been induced by a vacuum effect in the perilymph caused by the sudden drop in CSF pressure due to the dural rupture because of a patent cochlear aqueduct. It should be noted that this would have had to have been a bilateral disorder because the VEMP tests were abnormal bilaterally. It is a worthwhile suggestion by the reviewer to consider high resolution CT to look for patent cochlear aqueduct. Regrettably the authors omitted to mention a routine CT head scan undertaken in emergency at the time of the acute vertigo and headache. The scan was normal apart from the presence of a benign maxillary sinus retention cyst.
As with caloric testing, the writers do not undertake tests such as CT (which carries radiation effects) and MRI (because of high demand) unless the symptoms are severe enough to justify management change, which, in this case would have been to prevent a recurrence (highly unlikely) and would have required bilateral surgical intervention to occlude the cochlear aqueduct on both sides. However we give credit to the reviewer, as the authors did not think of this possibility.
The reviewer’s final suggestion of considering a central disorder as an explanation for the patient’s symptoms is highly unlikely in the writer’s opinion, as the third author, a neurologist with particular expertise in headache, could not find any neurological abnormality.
- Charles DA, Barber HO &Hope-Gill HF. Blood Glucose and Insulin Levels, Thyroid Function and Serology in Ménière’s Disease, Recurrent Vestibulopathy and Psychogenic Dizziness. J Otol Laryngol 1979;8:347-353.
- Lelièvre WC, Barber HO. Recurrent Vestibulopathy. Laryngoscope 1981;91;1-6.
- Rutka JA, Barber HO. Recurrent Vestibulopathy:Third Review. J Otolaryngol 1986;15:105-107.
- Longridge NS. Recurrent Vestibulopathy: Support for a viral etiology. J Otolaryngol 1989;18:99-100.
- Seemungal B, Rudge P, Davies R, Gresty M & Bronstein A. Three patients with migraine following caloric induced vestibular stimulation. J Neurol 2006;253:1000-1.
- Paparella MM, Mancini F. Trauma and Ménière’s Disease. Laryngoscope 1983;93:1004-1012.
Reviewer 2 Report
Comments and Suggestions for Authors
The Authors presented an unusual case involving the development of a bilateral abnormality of the otolith organs after an inadvertent dural puncture, hypothesizing that the vestibular damage resulted from either a mechanical effect of traction on the superior vestibular nerve or a vascular insult due to traction of vascular loops into the internal auditory canal, either of these possible mechanisms resulting in damage to the peripheral end organs bilaterally.
The case is indeed unusual, well presented and documented.
Author Response

(The authors gave the same response as above.)

Reviewer 3 Report
Comments and Suggestions for Authors
I am adding this new paper review. The authors study a [patient with a whiplash injury who underwent epidural BOTOX injections, after the third there was an inadvertent dural injury with low pressure headache and vertigo (no details of the exam then). Over time she developed PPPD, had a normal neuro-otology exam except for sway during Romberg and abnormal o' VEMP and later an abnormal c VEMP that was first normal.
I reviewed the citations of vertigo and low-pressure headache, the first resolved after a blood patch and the second had well documented Meniere's which could developed by changed in endolymphatic sac related to low CSF pressure. The same level of evidence is missing here. Suggest write as PPPD after low pressure injury, an unknown mechanism

Author Response

(The authors gave the same response as above.)
